# Fusing an agent-based model of mosquito population dynamics with a statistical reconstruction of spatio-temporal abundance patterns

**Sean M. Cavany**[1]*, **Guido España**[1], **Alun L. Lloyd**[2], **Gonzalo M. Vazquez-Prokopec**[3], **Helvio Astete**[4], **Lance A. Waller**[5], **Uriel Kitron**[3], **Thomas W. Scott**[6], **Amy C. Morrison**[7], **Robert C. Reiner, Jr**[8], **T. Alex Perkins**[1]*

**1** Department of Biological Sciences & Eck Institute of Global Health, University of Notre Dame, Notre Dame, Indiana, United States of America, **2** Department of Mathematics & Biomathematics Graduate Program, North Carolina State University, Raleigh, North Carolina, United States of America, **3** Department of Environmental Sciences, Emory University, Atlanta, Georgia, United States of America, **4** U.S. Naval Medical Research Unit Six, Lima, Peru, **5** Department of Biostatistics and Bioinformatics, Rollins School of Public Health, Emory University, Atlanta, Georgia, United States of America, **6** Department of Entomology and Nematology, University of California, Davis, Davis, California, United States of America, **7** Department of Pathology, Microbiology, and Immunology, School of Veterinary Medicine, University of California, Davis, Davis, California, United States of America, **8** Institute of Health Metrics and Evaluation, University of Washington, Seattle, Washington, United States of America

* sean.cavany@ndm.ox.ac.uk (SMC); taperkins@nd.edu (TAP)

**Data Availability Statement:** All computer code is available at github.com/scavany/mosquito_dynamics

## Abstract

The mosquito *Aedes aegypti* is the vector of a number of medically-important viruses, including dengue virus, yellow fever virus, chikungunya virus, and Zika virus, and as such vector control is a key approach to managing the diseases they cause. Understanding the impact of vector control on these diseases is aided by first understanding its impact on *Ae. aegypti* population dynamics. A number of detail-rich models have been developed to couple the dynamics of the immature and adult stages of *Ae. aegypti*. The numerous assumptions of these models enable them to realistically characterize impacts of mosquito control, but they also constrain the ability of such models to reproduce empirical patterns that do not conform to the models' behavior. In contrast, statistical models afford sufficient flexibility to extract nuanced signals from noisy data, yet they have limited ability to make predictions about impacts of mosquito control on disease caused by pathogens that the mosquitoes transmit without extensive data on mosquitoes and disease. Here, we demonstrate how the differing strengths of mechanistic realism and statistical flexibility can be fused into a single model. Our analysis utilizes data from 176,352 household-level *Ae. aegypti* aspirator collections conducted during 1999–2011 in Iquitos, Peru. The key step in our approach is to calibrate a single parameter of the model to spatio-temporal abundance patterns predicted by a generalized additive model (GAM). In effect, this calibrated parameter absorbs residual variation in the abundance time-series not captured by other features of the mechanistic model. We then used this calibrated parameter and the literature-derived parameters in the agent-based model to explore *Ae. aegypti* population dynamics and the impact of insecticide spraying to kill adult mosquitoes. The baseline abundance predicted by the agent-based

**Funding:** SMC, GFCE, GMVP, ACM, TWS, RCR, and TAP were supported by grant P01AI098670 (TWS, PI) from the National Institutes of Health, National Institute for Allergy and Infectious Disease (https://www.niaid.nih.gov). In addition, this work was supported by the NIH National Institute of General Medical Sciences R35 MIRA program to TAP (R35GM143029). The funders had no role in the study design, data collection and analysis, decision to publish, or preparation of the manuscript.

**Competing interests:** The authors have declared that no competing interests exist.

model closely matched that predicted by the GAM. Following spraying, the agent-based model predicted that mosquito abundance rebounds within about two months, commensurate with recent experimental data from Iquitos. Our approach was able to accurately reproduce abundance patterns in Iquitos and produce a realistic response to adulticide spraying, while retaining sufficient flexibility to be applied across a range of settings.

## Author summary

The mosquito *Aedes aegypti* is the vector for a number of the most medically important viruses, including dengue, Zika, chikungunya, and yellow fever. Understanding the population dynamics of this mosquito, and how those dynamics might respond to vector control interventions, is critical to inform the deployment of such interventions. One of the best ways to gain this understanding is through modeling of population dynamics. Such models are often categorized as either statistical or dynamical, and each of these approaches has advantages and disadvantages–for instance, statistical models may more closely match patterns observed in empirical data, while dynamical models are better able to predict the impact of counterfactual situations such as vector control strategies. In this paper, we present an approach which fuses these two approaches in order to gain the advantages of both: it fits empirical data on *Aedes aegypti* population dynamics well, while producing realistic responses to vector control interventions. Our approach has the potential to inform and improve the deployment of vector control interventions, and, when used in concert with an epidemiological model, to help reduce the burden of the diseases spread by such vectors.

## Introduction

The mosquito *Aedes aegypti* is found throughout tropical and subtropical regions of the world and on all continents except Antarctica [1]. As the main vector of important viruses–namely, dengue, yellow fever, chikungunya, and Zika viruses–its large and expanding range is a cause for concern [1]. Dengue virus alone causes 10,000 deaths each year [2]. Though there is now a vaccine for dengue, it leads to an increased risk of severe disease in individuals without previous exposure [3], and there are no currently-licensed vaccines for chikungunya or Zika [4–6]. Hence, in many settings, the only intervention available to counter these viruses is mosquito vector control. Accurately estimating how *Ae. aegypti* populations will respond to vector control will help inform optimization of strategies for its deployment.

While the link between vector indices and dengue virus (DENV) transmission is complicated, vector density is an important driver of transmission of *Aedes*-transmitted viruses [7–9]. *Ae. aegypti* population dynamics are driven by a range of environmental factors, including temperature, sunlight, humidity, and rainfall [10]. Eggs need water and hospitable temperatures to hatch, but can survive long periods without desiccating, of six months or more [11]. Larval and pupal development and mortality are dependent on water temperature, which depends on the amount of direct sunlight they receive and the ambient air temperature [10,12–14]. Adult mosquitoes' egg-laying cycles and mortality rates are also strongly temperature dependent. Density-dependent larval mortality, due to competition for limited resources, plays an important role in modulating population sizes [15–17]. Mortality can also occur during the larval and pupal stages as containers are cleaned or accidentally spilled [18].

Several previous studies attempted to generate statistical estimates of mosquito abundance [19,20], typically based on trap data or mark-release-recapture experiments [21]. The flexibility of these models facilitates the capture of signals from noisy experimental data, but these models encounter difficulty when attempting to predict the impact of control strategies not included in the data, because they do not contain a mechanistic understanding of population dynamics and the associated non-linearities. Other efforts have focused on predicting the range and potential distribution of *Ae. aegypti*, including under climate change [1,22]. Such models do not predict local abundance or fine-scale temporal and spatial heterogeneities, and so, while important for understanding broader patterns and changes, they are not appropriate for modeling vector control, for which individual households are the most appropriate scale [9].

Two of the most significant mechanistic models of *Ae. aegypti* population dynamics are AedesBA, developed in Otero et al. [23], and Skeeter-Buster, developed in Magori et al. [10] based on CiMSiM by Focks et al. [12,13]. AedesBA is a stochastic compartmental model that considers three immature stages (eggs, larvae, and pupae) and three adult stages. The model is spatially-explicit and the geographic units are patches on a grid. Density-dependent mortality occurs directly in the larval state at a rate that is a quadratic function of the number of larvae. Skeeter-Buster models individual containers in which immature stages develop. Individual larvae and pupae develop after reaching body-size thresholds, and grow at rates dependent on resource availability and temperature. Larval density dependence is modeled indirectly by reduced food availability. Mechanistic models such as these are able to make realistic projections of the impact of vector control, but can struggle to match patterns of abundance that do not closely conform to the models' typical behavior (e.g., see the fall in Fig 3 of [24]).

While both statistical and mechanistic models have their advantages, these different modeling frameworks are typically applied separately. Unifying these two approaches would enable the combined model to (1) accurately recreate observed patterns of abundance through its statistical component and (2) explore how population patterns could have differed under counterfactual situations through its mechanistic component. Moreover, the unified model could be used to explore the interaction between seasonality and how populations respond to forced perturbations like vector control. One of the main challenges is ensuring that the dynamics of the mechanistic model are able to match those of the statistical model. In this study we demonstrate an approach that fuses the differing strengths of mechanistic realism and statistical flexibility into a single model. This method enabled us to accurately recreate temporal patterns of abundance, and to predict the impact of vector control interventions on abundance. Our model is sufficiently detailed to recreate spatial heterogeneity, allowing analysis of spatially-targeted interventions.

## Methods

### Ethics statement

The study protocol was approved by the Naval Medical Research Unit No. 6 (NAMRU-6) Institutional Review Board (IRB) (protocol #NAMRU6.2014.0028), in compliance with all applicable Federal regulations governing the protection of human subjects. IRB relying agreements were established between NAMRU-6, the University of California, Davis, Tulane University, Emory University and Notre Dame University. The protocol was reviewed and approved by the Loreto Regional Health Department, which oversees health research in Iquitos. This study represents historical data analysis using data without personal identifiers.

## Outline of approach

First, we took statistically-derived estimates of mosquito abundance over the period 2000–2010 for the city of Iquitos, Peru [19]. These estimates were derived from a negative binomial generalized additive model (GAM) based on mosquito abundance surveys from Iquitos. Model predictions varied in both space and time, but, in the absence of intervention, not space and time together. This model was thoroughly described by Reiner et al. [19].

1. Our second step used a deterministic ordinary-differential-equation (ODE) model of *Aedes aegypti* population dynamics to derive a mortality rate time series. The model combined predictions from Step 1 with temperature data from Iquitos and established relationships between model parameters and temperature. More details are given below in the Deterministic model section.

2. In the final step we used the time-series derived in Step 2 to drive the dynamics of a stochastic, agent-based model. This enabled the baseline dynamics of the agent-based model to closely match those predicted by the statistical model in Step 1. More details on the agent-based model (ABM) are given below in the Agent-based model section.

   These steps are summarized in Fig 1.

## Deterministic model

To link the statistical model and the ABM, we used an ODE model as an intermediate step, described in Eqs 1–4:

$$\frac{dE}{dt}(x, t) = \frac{n_E a(T(t))N(x, t)}{2} - (d_E(T_W(t)) + \mu_E(T_{W,max}(t)))E(x, t) \qquad 1$$

$$\frac{dL}{dt}(x, t) = d_E(T_W(t))E(x, t) - \left(d_L(T_W(t)) + \mu_L(T_{W,max}(t)) + \mu_c(t) + \frac{L^2(x, t)}{\kappa(x)}\right)L(x, t) \qquad 2$$

$$\frac{dP}{dt}(x, t) = d_L(T_W(t))L(x, t) - \left(d_P(T_W(t)) + \mu_P(T_{W,max}(t)) + \mu_c(t)\right)P(x, t) \qquad 3$$

$$\frac{dN}{dt}(x, t) = d_P(T_W(t))P(x, t) - \mu_N(T_{max}(t))N(x, t), \qquad 4$$

where $E$ is the number of eggs, $L$ is the number of larvae, $P$ is the number of pupae, and $N$ is the number of female adults. The parameters are described in Table 1.

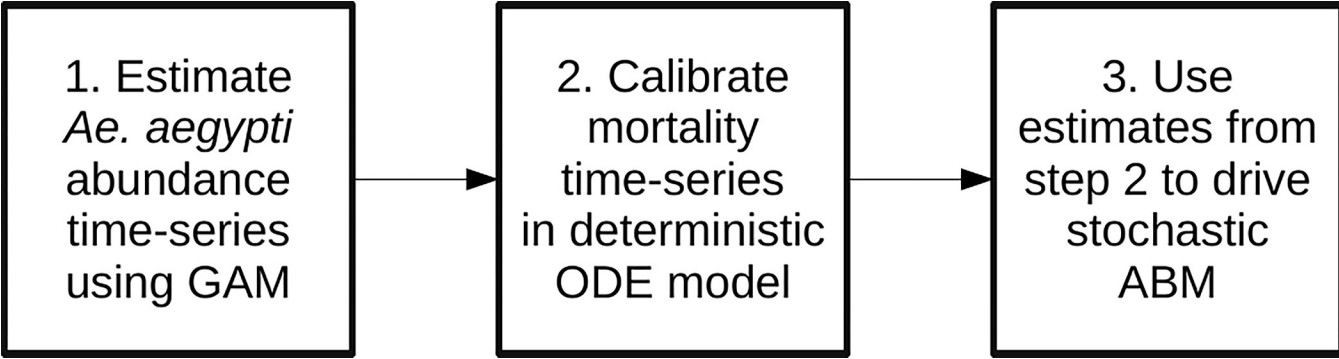

**Fig 1. Flow chart of steps in methodology.** GAM = Generalized additive model; ODE = Ordinary differential equations; ABM = Agent-based model.

**Table 1. Parameter names, definitions, and sources.**

| Symbol | Definition | Source |
|--------|-----------|--------|
| $n_E$ | Number of eggs laid per gonotropic cycle | Otero et al. [23] |
| $a$ | Gonotrophic-cycle rate | Magori et al. [10] |
| $d_E$ | Development rate of eggs | Magori et al. [10] |
| $d_L$ | Development rate of larvae | Magori et al. [10] |
| $d_P$ | Development rate of pupae | Magori et al. [10] |
| $\mu_E$ | Mortality rate of eggs | Magori et al. [10] |
| $\mu_L$ | Mortality rate of larvae | Magori et al. [10] |
| $\mu_P$ | Mortality rate of pupae | Magori et al. [10] |
| $\mu_N$ | Mortality rate of female adults | Magori et al. [10] |
| $\mu_C$ | Other sources of larval and pupal mortality rate | Estimated by fitting deterministic model to estimated larval abundance in Iquitos. Used to fuse the deterministic and statistical models. |
| $\kappa$ | Carrying capacity of larvae | Estimated so that density-dependence contributes 90% of deaths on average. |

The gonotrophic cycle rate parameter ($a(T)$), development rate parameters (($d_E(T_W)$, $d_L(T_W)$, $d_P(T_W)$)), and mortality rate parameters ($\mu_E(T_{W,max})$, $\mu_L(T_{W,max})$, $\mu_P(T_{W,max})$, $\mu_N(T_{max})$)) were driven by daily data on mean, minimum, and maximum temperatures in Iquitos between 2000–10. Formulae describing these parameters were obtained from Otero et al. and Magori et al. [10,23]. Although the expressions describing these parameters are functions of both daily minimum and maximum temperatures (S1 Text), in the period 2000–2010 the daily minimum temperature in Iquitos never went below the threshold at which minimum temperature affected mortality, so we write these parameters as functions of $T_{max}$ only. Temperature parameters with the subscript $W$ refer to water temperature, derived from formulae relating this to mean, minimum, and maximum air temperature (S1 Text), and assuming an average of 10% sun exposure. Temperature data was used at the city level, so each of these parameters varies in time, but not space. The number of eggs laid per gonotrophic cycle, $n_E$ was fixed at 63, as in Otero et al. [23]. While there is variability in the number of eggs laid that is unrepresented here, some of that variability will be approximated by variability in the gonotrophic cycle length. We modeled density dependent larval mortality as proportional to the square of the number of larvae [15,16]. Noting that mortality in the larval stage is dominated by density dependent effects [17,18,25], the carrying capacity, $\kappa(x)$, was chosen so that density-dependent mortality is an order of magnitude greater than natural density-independent mortality at average larval densities; i.e.,

$$10\overline{\mu_L(t)} = \frac{\overline{L(x,t)}}{\kappa(x)}.$$

The carrying capacity, $\kappa(x)$, is location-specific and varies in space but not in time. The term $\mu_c(t)$ represents additional larval and pupal mortality not accounted for otherwise. This could be caused in part by changes in precipitation or by routine container maintenance and cleaning, a source of mortality for immature stages that is not easily quantified [18]. Calculation of the time-varying parameter $\mu_c(t)$ is the key step in our approach, because it is calibrated to spatiotemporal estimates of mosquito abundance in Iquitos during 2000–10 by Reiner et al. [19], and it enabled us to account for differences between those estimates and the ODE. To calibrate this parameter, we took the following steps.

- First, we used Eq 4 alongside estimates of $N(t)$ from the statistical model, to obtain estimates of $P(t)$. In this and future steps we found the derivative of a time series (e.g., $dN/dt$) by taking centered differences (i.e., $\frac{dX}{dt} \simeq \frac{X(t+1)-X(t-1)}{2}$, where $X$ is either $E$, $L$, $P$, or $N$, and $t$ is measured in days) at all time points except the first and last, at which we took forward (i.e., $\frac{dX}{dt} \simeq \frac{X(1)-X(0)}{1}$) and backward (i.e., $\frac{dX}{dt} \simeq \frac{X(n)-X(n-1)}{1}$, where n is the final day of the time series) differences respectively. We could then estimate $P(t)$ from

$$P(x,t) = \frac{\left(\frac{dN}{dt}(x,t) + \mu_N(T_{max}(t))N(x,t)\right)}{d_P(T_W(t))}.$$

- Second, we obtained $E(t)$ by integrating Eq 1 using the deSolve package in R. For this and all other integrations we used the radau method, with tolerances kept at their default values of $1 \times 10^{-6}$.

- Third, we obtained $L(t)$ by combining Eqs 2 and 3 to remove $\mu_c$ from them, yielding a first-order differential equation in L, which we solve numerically using deSolve:

$$L(x,t) = d_E E + \left(\frac{1}{P}\frac{dP}{dt} + d_P + \mu_P - d_L - \mu_L\right)L - \frac{d_L L^2}{P} - \frac{L^3}{\kappa}.$$

- Finally, we obtained $\mu_c(t)$ by rearranging Eq 2 to obtain

$$\mu_c(t) = \frac{\left(d_E(T_W(t))E(x,t) - \frac{dL}{dt}(x,t) + \left(d_L(T_W(t)) + \mu_L(T_{W,max}(t)) + \frac{L^2(x,t)}{\kappa(x)}\right)L(x,t)\right)}{L(x,t)}.$$

We insist that all parameters remain non-negative at all time-steps. Although Eqs 1–4 contain a spatial component, we only needed to undertake the process once for the normalized system (i.e., the one in which $N(x, 0) = 1$). The appropriate larval capacity for each location can be obtained by scaling $\kappa(x)$ by $N^2(x, 0)$. The resulting time series for $\mu_c(t)$ is shown in S1 Fig.

## Agent-based model

We incorporated the $\mu_c(t)$ time series obtained from manipulation of the ODE model into an ABM of DENV transmission based on the one previously used in Perkins et al. [26]. This model contains location data for 92,891 buildings in Iquitos. The model contains modules for human agents and virus transmission by *Ae. aegypti*. These human and virus transmission components were not used in the present study. At each location, we modeled the number of eggs, larvae, pupae, and adult female mosquitoes; only female *Ae. aegypti* bite humans and transmit virus. The number of adult mosquitoes that emerge at a given location, $x$, on a given day, $t$, is modeled as a Poisson random variable, with rate parameter given by $d_P(T)P(x, t)$, where $x$ is the house location, and $t$ is the day. Adult mosquitoes die at a daily rate $\mu_N(T)$. The number of immature mosquitoes and eggs are modeled by integrating Eqs 1–3; i.e., all developmental transitions in the immature stages are modeled deterministically. The $\mu_c(t)$ time-series forces the baseline dynamics, in the absence of interventions, to closely match the abundance predicted by the statistical model. The location-specific carrying capacity, $\kappa(x)$, maintains the spatial heterogeneity estimated in Reiner et al. [19], and also determines the

equilibrium population density to which the population returns following spraying. Mosquitoes of all stages only exist in the model within buildings. Adult mosquitoes move to a location within 100 m of their current location with daily probability of 0.3 [10]. Locations within this radius are chosen with equal probability. Each of the main text plots showing the mosquito time series output from the ABM show a single simulation. In this model, the mosquito population dynamics at the city-level do not appear very stochastic. This is likely because of the large population size ($\sim10^5$), the deterministic treatment of the immature stages, and the fact that the time series is strongly forced by $\mu_c$. We show the lack of variability across 400 simulations of the ABM in S2 Fig. We also include text describing the ABM in more detail in S2 Text.

## Experiments

We examined the effect of spraying with insecticide via numerical simulation, with either an instantaneous effect (ultra-low volume spraying; ULV) or a residual effect (targeted insecticide residual spraying; TIRS). The former increased the adult mortality rate on the day of spraying by 1.5 deaths/day, calibrated so that abundance following spraying was 60% of baseline, as observed in an empirical study of an actual city-wide spraying campaign in Iquitos in 2014 [27] (S3 Fig). There was no effect on the mortality of the immature stages. In the agent-based model, the daily mortality rate is converted into a daily probability of death according to $prob = 1-exp(-rate)$. The city-wide ULV campaign typically took around 27 days to complete, and consisted of three rounds during which each was sprayed once with probability 0.7. The length of the campaign and the probability that a house was sprayed in a given round were chosen to reflect past ULV spraying campaigns in Iquitos, Peru, in which an average of 11,000 houses are sprayed per day. TIRS increased the adult mortality rate by 9 deaths/day and lasted for 90 days, after which the effect decayed exponentially [28]. As for ULV, there was no effect on the mortality of the immature stages. City-wide TIRS campaigns took around 39 days to complete, and consisted of just one round, with the same probability of 0.7 that an individual house is sprayed. This length was based on the observation that TIRS takes ~5 times as long to apply as ULV [29], and amounted to 2,000 houses being sprayed per day. For both types of insecticide application, we examined the effect on total abundance over time, spatial heterogeneity in abundance, and the female adult mosquito age distribution. Zones were sprayed in ascending order according to the numbers displayed in Fig 2.

## Results

### Calibration

The total predicted abundance of female adults in both the ODE model and the ABM closely matched the total abundance predicted by the GAM (Fig 3). Here the ODE model results were obtained by integrating Eqs 1–4 using the derived $\mu_c(t)$. The root mean squared deviation between the GAM and the calibrated ODE model was 43,200 and between the GAM and the ABM it was 48,000. The ODE model does not match the GAM perfectly due to the fact that all parameters were forced to be non-negative and the discretization of the ODE system, both of which cause small discrepancies to be introduced. Thus, the ABM matched the GAM nearly as well as the calibrated ODE model. The Pearson's correlation coefficients of the ABM and the ODE model with the GAM were 0.946 and 0.950 respectively, and the correlation coefficient of the ABM with the ODE model was 0.997. When abundances were low, however, the ABM gave slightly lower predictions than those of the additive model. This was likely due to the presence of demographic stochasticity in the ABM. Both mechanistic models (i.e. ODE and ABM) smoothed out some of the noise of the GAM prediction, leading to smaller peaks and reduced day-to-day oscillations (Fig 3).

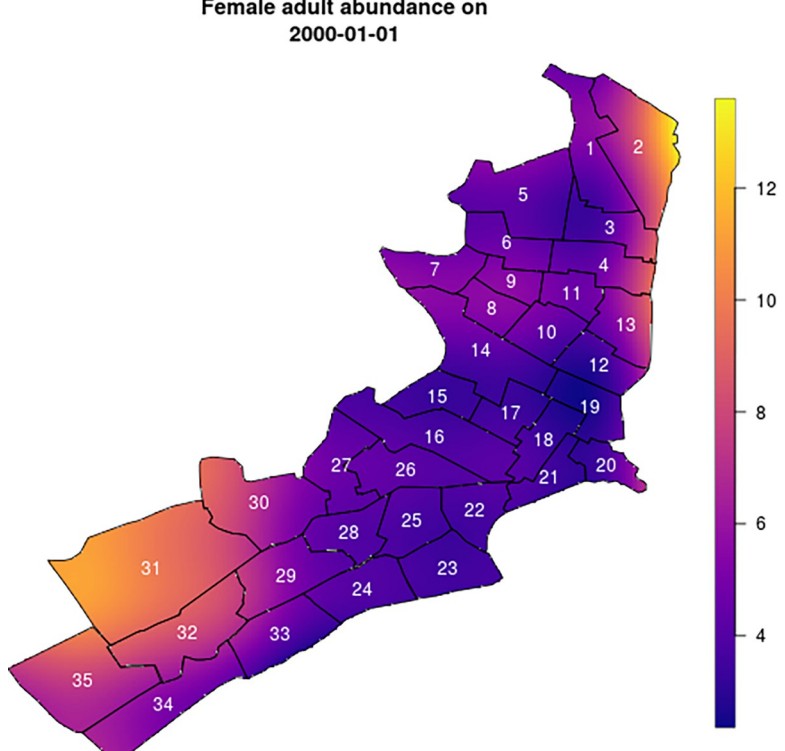

**Fig 2. Smoothed map of adult female abundance on 1ˢᵗ January 2000 (the start of the simulation).** Warmer colors (in the Southwest and Northeast of the city) indicate regions of highest abundance. The color scale represents the average number of mosquitoes per household. The shape files for the underlying maps can be found at github.com/scavany/mosquito_dynamics.

Spatial heterogeneity in the ABM is determined by the location-specific larval carrying capacity. This maintained the same spatial patterns over time as predicted by the GAM (Figs 2 and 4). Fig 2 shows the initial spatial distribution of female adult mosquito abundance in the

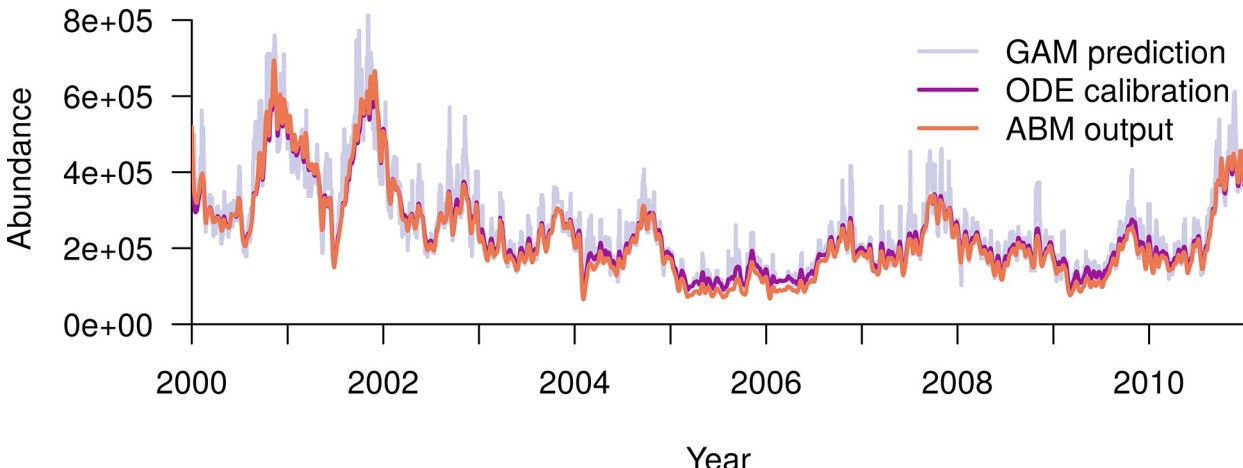

**Fig 3. The total predicted city-wide abundance of female adult Ae. aegypti is very similar across the three models.** The periwinkle blue line shows daily values of abundance predicted by the GAM, the purple line shows those predicted by the ODE model, and the pink line those predicted by the ABM. GAM: generalized additive model; ODE: ordinary differential equation model; ABM: agent-based model.

## Proportion of female adults in each zone

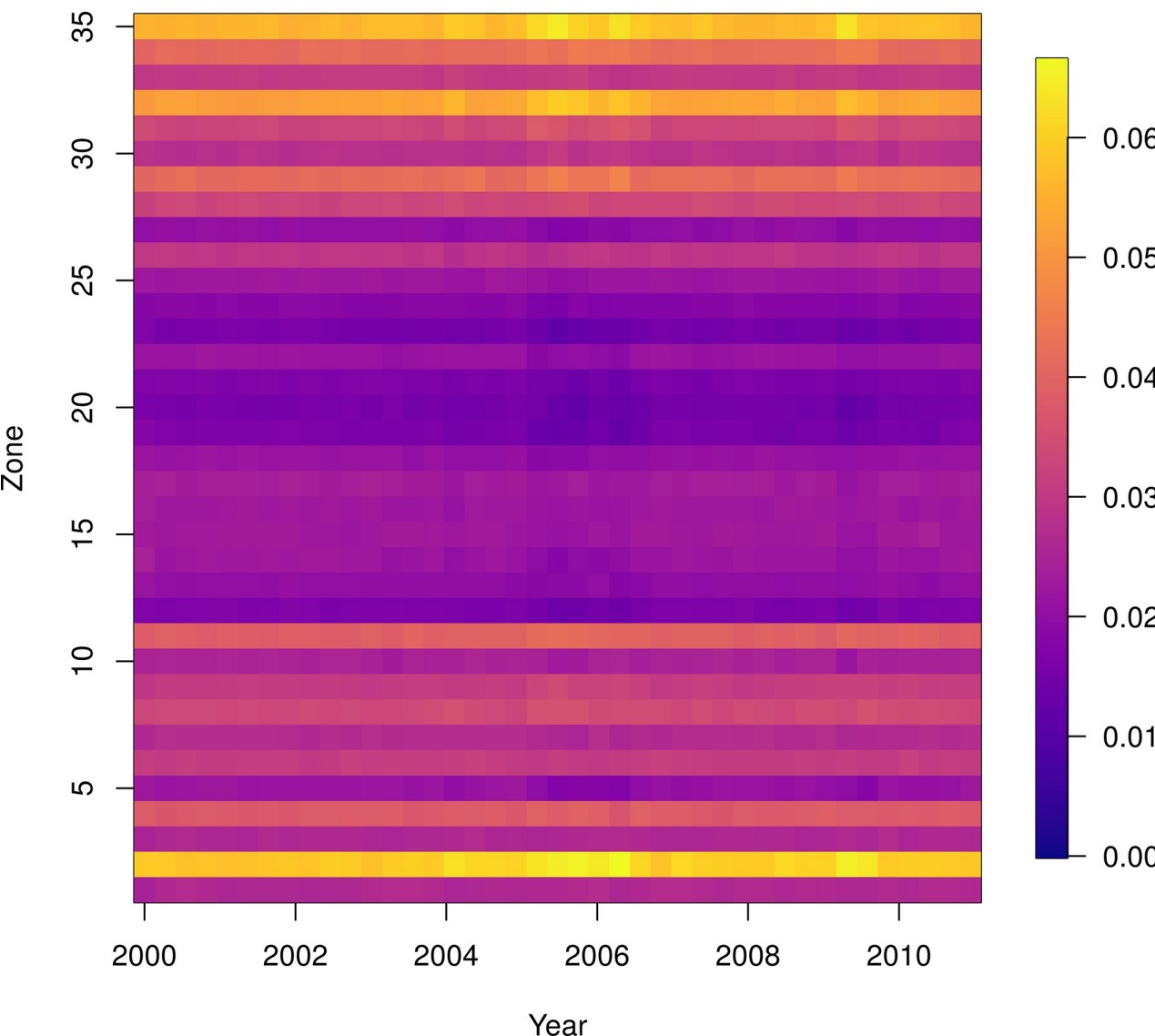

**Fig 4. Normalized female adult abundance over time under the agent-based model: the relative abundance across zones does not substantially vary through time.** Each column represents the daily abundance every 100 days from 2000–2010. Each row is a Ministry of Health zone in Iquitos. Columns are normalized by the total female adult abundance across all zones that day, so each column sums to one and the color represents the proportion of mosquitoes in that zone that day.

agent-based model, though by definition this is identical across the three models. Fig 4 shows the normalized abundance in each of 35 Ministry of Health (MoH) zones every hundred days, where the normalizing factor for each day was the total number of adult females across all zones on that day (S4 Fig shows the absolute values of the same output). The lack of variation over time in each zone indicates that relative abundance and spatial heterogeneity within each zone are relatively constant. This is what we would expect because in the ODE model and the GAM the relative abundance between zones is constant through time. The small changes in

relative abundance of adult females apparent in the agent-based model (Fig 4) are due to stochastic fluctuations in local abundance.

## Experiments

**Spatiotemporal effects of spraying.** Following a city-wide ULV campaign, the ABM-predicted abundance rebounded to within 10% of its baseline level 2.1 months after the start of the campaign, and to within 1% of baseline after 3.7 months (Fig 5). Female adult abundance reached a minimum of 129,000, which was 60% less than the abundance in the absence of spraying on the same date (317,000). It reached its minimum 28 days after the start of spraying, and the whole campaign took 27 days. Following a city-wide TIRS campaign, the total abundance was maintained at a very low level (<10% of baseline) for six months, reaching a minimum of 1,250 mosquitoes 5.2 months after beginning spraying. The TIRS campaign took 39 days. Abundance eventually returned to within 10% of baseline after 10 months, and to within 1% after a year.

Following the ULV campaign, there was very little discernible effect on spatial patterns of mosquito abundance (Fig 6; total abundance by zone shown in S5 Fig). Following the TIRS campaign, however, there was a larger effect on the spatial patterns. MoH Zone 2, which had the highest abundance at baseline, had a proportionally even greater abundance than all other zones following TIRS spraying (Fig 7 and S1 Video; total abundance by zone shown in S6 Fig). Note that abundance drops substantially in all zones following TIRS, and the yellow region in Fig 7 reflects the fact that a much greater proportion of the total mosquitoes are in Zone 2 following spraying. This is in part because *Ae. aegypti* were eliminated from many buildings in the city in our simulations, such that in some zones there were none remaining. Because Zone 2 had much higher baseline abundance, the probability of local extinction was much lower and more buildings continued to contain *Ae. aegypti*. To examine whether lower baseline abundance can explain these differences in abundance following spraying, we set the baseline abundance in every location to be equal to that of the location with highest abundance. Following spraying in that scenario, we saw a less stark difference between the zones, but nonetheless the pattern remained (S7 Fig). We also explored the effect of spraying with hypothetical insecticides that had (i) a small effect on mortality (i.e., equal to ULV) but a long residual effect, and

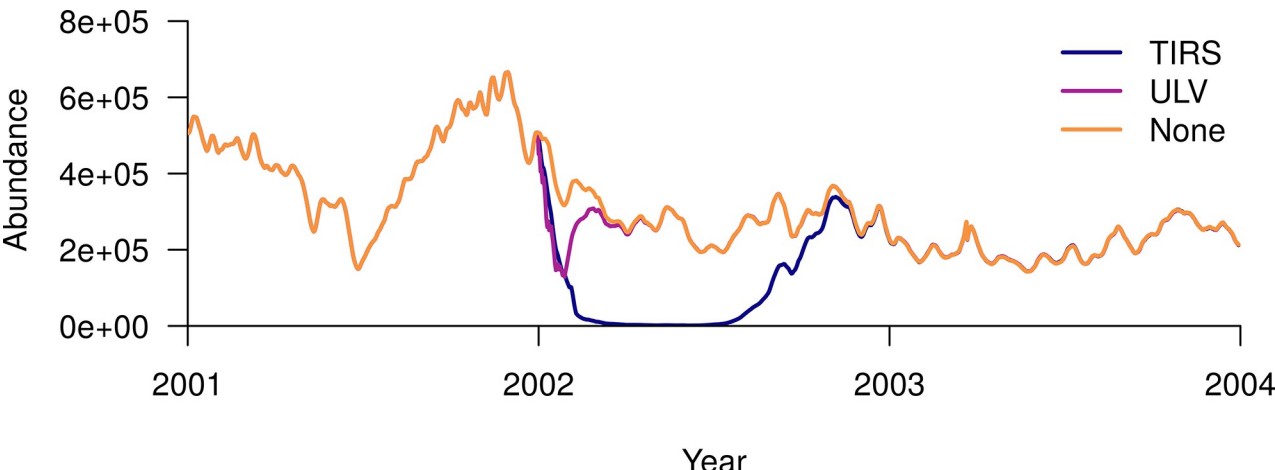

**Fig 5. Response of female adult abundance to a city-wide residual insecticide campaign (TIRS) and a city-wide campaign with no residuality (ULV), both initiated on the 1st of January 2002.** The orange line shows the baseline scenario of no-spraying, the purple line the ULV scenario, and blue line the TIRS scenario.

## Proportion of female adults in each zone following ULV

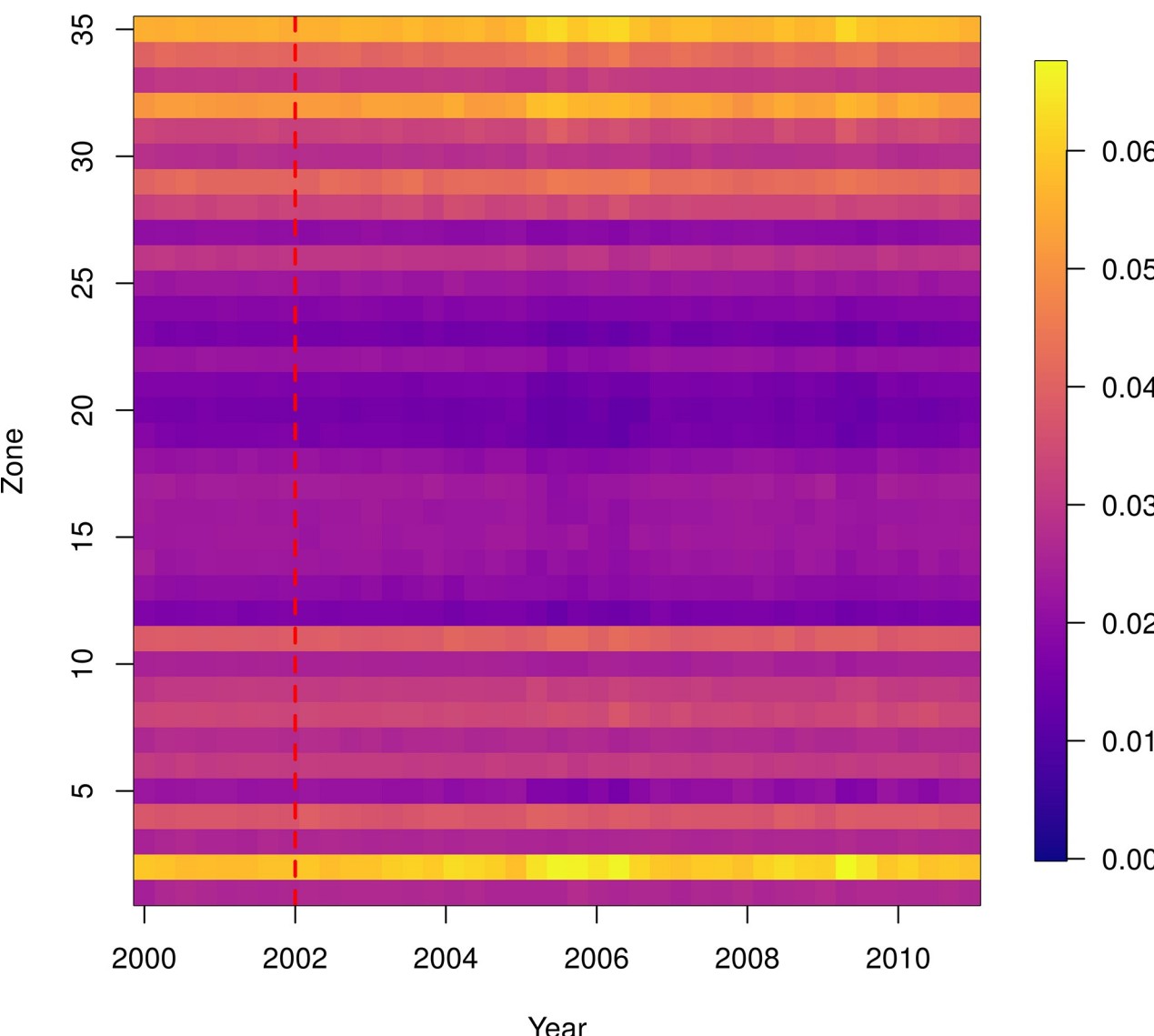

**Fig 6. Normalized female adult abundance over time, following a city-wide ULV campaign initiated on the 1<sup>st</sup> January 2002, indicated by the red dashed line.** ULV spraying does not have a discernable effect on the spatial distribution of abundance. Each column represents the daily abundance every 100 days from 2000–2010. Each row is a Ministry of Health zone in Iquitos. Columns are normalized by the total female adult abundance across all zones that day, so each column sums to one and the color represents the proportion of all female adult mosquitoes in that zone that day.

(ii) a large effect on mortality (i.e., equal to TIRS) but no residuality (S8 Fig). Scenario (i) (low increase in mortality, high residuality) produced a similar pattern of abundance to that of the TIRS campaign (Fig 7) and scenario (ii) (high increase in mortality, low residuality) produced a pattern similar to the ULV campaign. This suggests that the residual effect of TIRS is more important to its improved overall impact compared to ULV than its larger baseline effect on mortality.

**Mosquito age distribution following spraying.** In the absence of spraying, mosquito ages were roughly exponentially distributed (Fig 8 and S2 Video). Through most of the simulation,

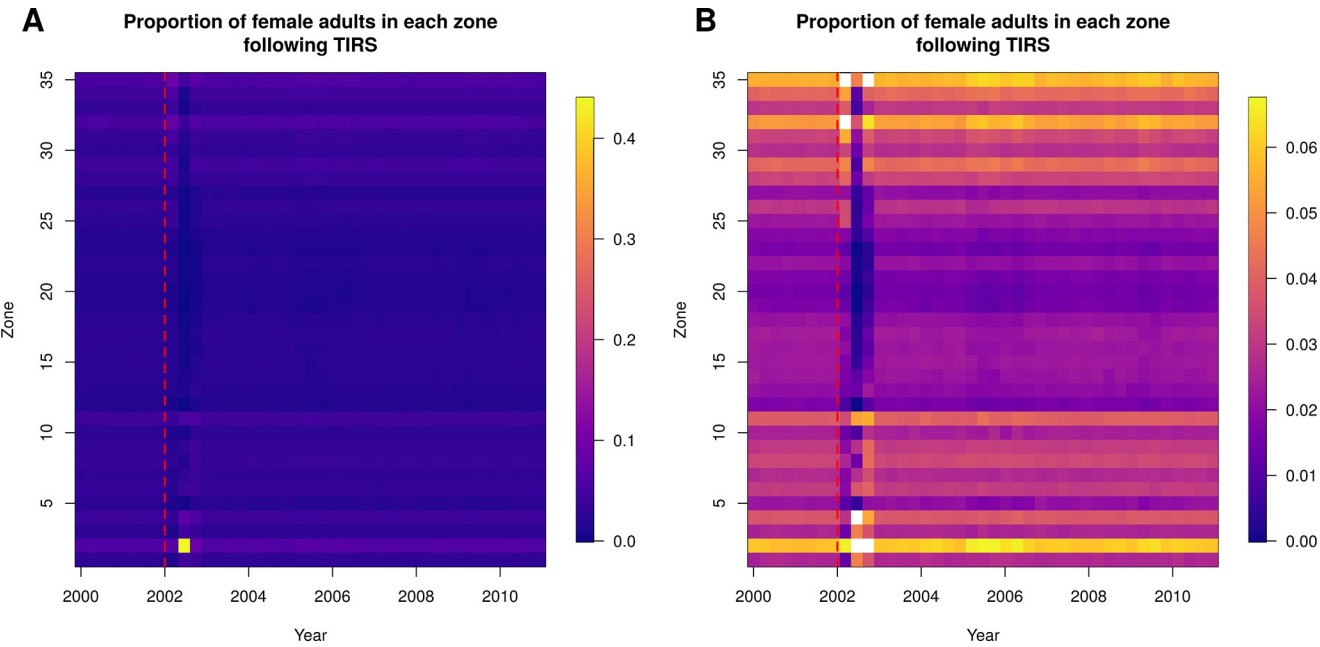

**Fig 7. Normalized female adult abundance over time, following a city-wide TIRS campaign initiated on the 1st January 2002, indicated by the red dashed line.** A. TIRS has a substantial effect on the spatial distribution of abundance for most of the year following spraying, as some zones are reduced to zero abundance. Each column represents the daily abundance every 100 days from 2000–2010. Each row is a Ministry of Health zone in Iquitos. Columns are normalized by the total female adult abundance across all zones that day, so each column sums to one and the color represents the proportion of mosquitoes in that zone that day. B. As in A, but with the same color-scale as Fig 4.

there were many more young mosquitoes and few very old mosquitoes. Occasionally, such as near the start of 2004, a cohort of adult mosquitoes survived longer and the age distribution became less skewed and sometimes bimodal (around the 1 minute mark in S2 Video; also see S9 and S10 Figs). This is likely a consequence of the precipitous drop in abundance around this time necessitating a large value of $\mu_c(t)$. This abrupt drop in abundance could be due to a physical event, for instance a larval habitat reduction campaign or a flooding event flushing out containers, or it could be an artifact of the statistical model fit. This in turn leads to a temporarily large drop in the total population of larvae and pupae and hence a 'missing' cohort of adult mosquitoes and a bimodal age distribution.

Following city-wide ULV and TIRS campaigns, the average age of the mosquito population was reduced, with proportionally more young mosquitoes (Figs 9 and 10 and S3 and S4 Videos). In the case of TIRS, this effect was stronger and more long-lasting.

## Discussion

The principal contribution of this study is the development of a novel framework for fusing a flexible statistical model with a detailed mechanistic model. The underlying statistical model allowed us to accurately capture observed empirical patterns. This was linked to an ABM, which was able to recreate the spatio-temporal patterns predicted by the statistical model. This approach retains the flexibility of a mechanistic model, allowing us to look at dynamic effects of perturbations to the population, including capturing feedbacks caused by population-dynamic effects. We demonstrated this using city-wide insecticide spraying campaigns as a case study. Such campaigns only had an effect on predicted patterns of spatial heterogeneity if TIRS was used, in which case the higher efficacy of this treatment led to many buildings experiencing local extinction of *Aedes aegypti*, drastically changing the spatial patterns of

## Proportion of female adults at each age

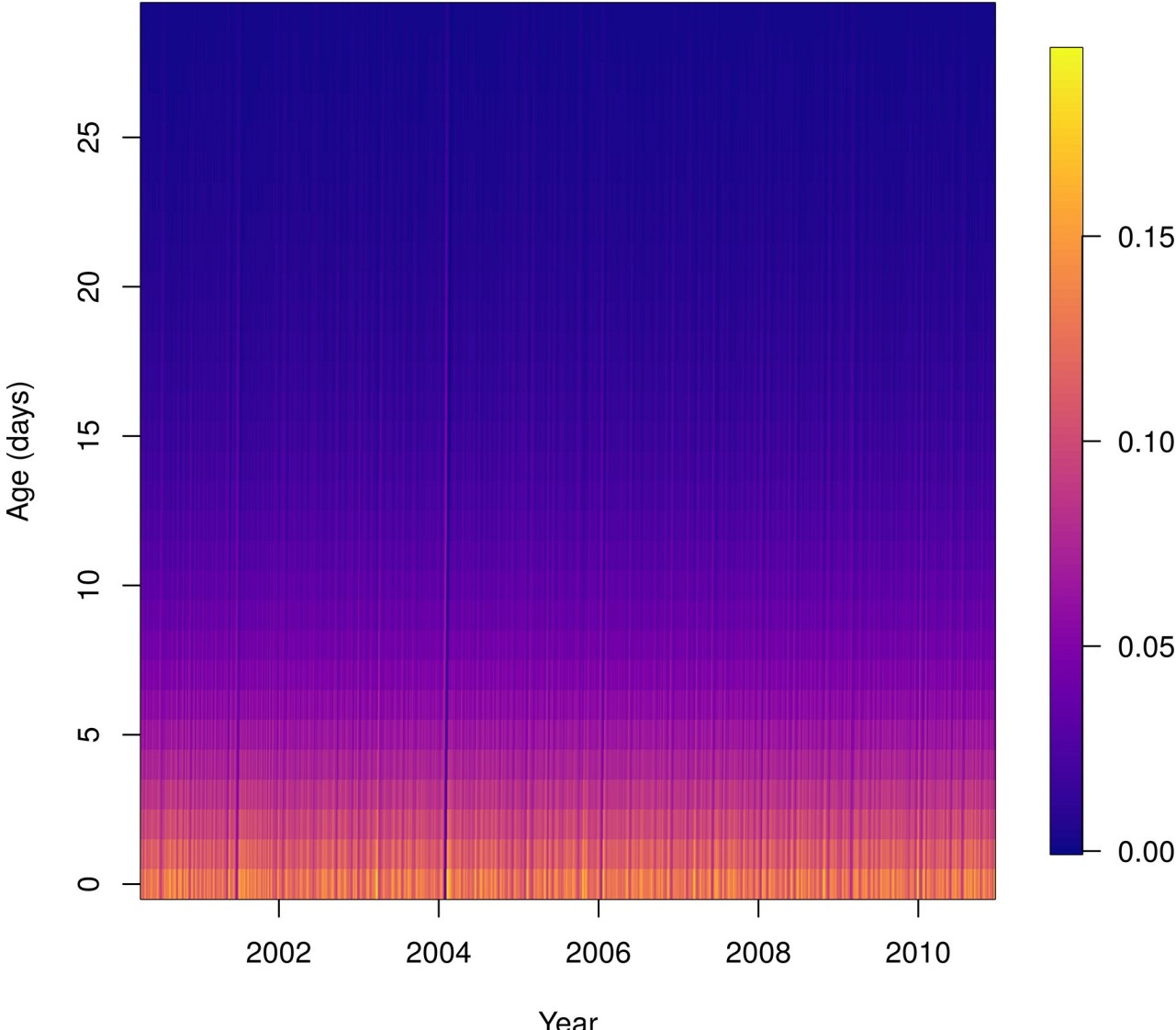

**Fig 8. Normalized female adult age distribution over time: mosquito ages were roughly exponentially distributed.** Each column shows the daily proportion of mosquitoes of the age shown on the y-axis. Warmer colors represent higher proportions.

female adult abundance. Whether TIRS or ULV was used, the average age of mosquitoes was much lower following the campaign, but with TIRS that effect lasted much longer.

Our framework enabled the ABM to capture the patterns predicted by the statistical model very well. We were able to capture both the spatial pattern and the majority of the temporal pattern in mosquito abundance, with two slight exceptions. The first exception was when total mosquito abundance fell to very low levels in 2005–06, in which case the ABM slightly under-predicted abundance. This was likely due to the fact that, when abundance was low, there was a greater proportion of houses that contained zero mosquitoes, due to demographic stochasti-city captured by the discrete population sizes. This effect cannot happen in the ODE, which

## Difference in proportion of female adults in each zone following ULV

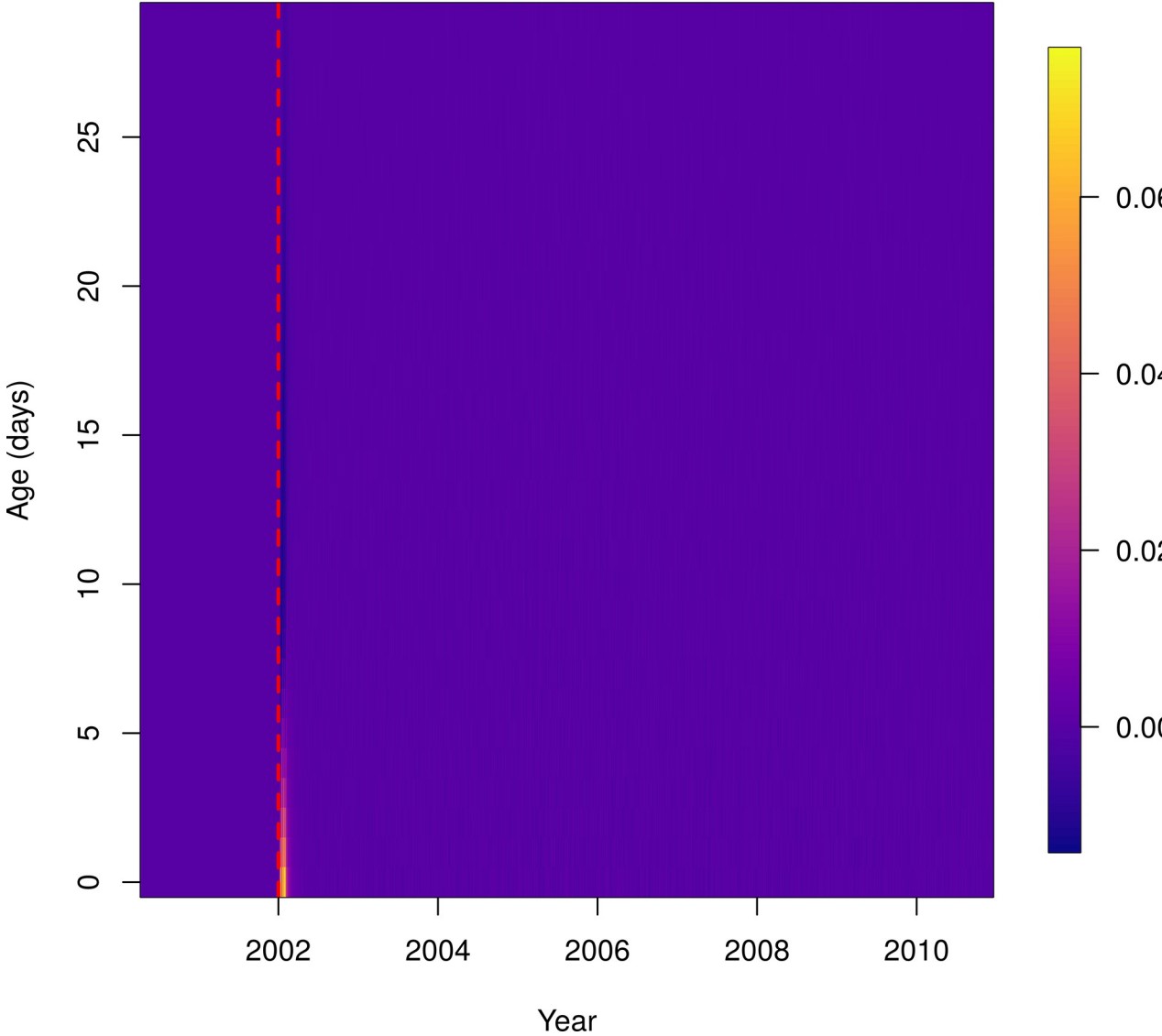

**Fig 9. Normalized difference in female adult ages over time between baseline and a city-wide ULV campaign initiated on 1ˢᵗ January 2002, indicated by the red dashed line.** Following spraying there are proportionally more young Ae. aegypti. Each column shows the difference in the daily proportion of mosquitoes of a given age, shown on the y-axis. Warmer colors represent higher proportions.

models the dynamics of abundance deterministically. This same effect also likely forms part of the explanation for the pattern of spatial heterogeneity observed following the TIRS campaign; i.e., the much higher relative abundance in Zone 2 compared to all other zones. Following TIRS, mosquito abundances city-wide were reduced to very low levels, leading to many locations with no mosquitoes. The higher baseline in Zone 2 enables more locations to avoid this stochastic effect, and hence a proportionally greater abundance. The second exception is that both the ODE model and the ABM smooth out some of the day-to-day variability in mosquito abundance predicted by the GAM. This is likely a consequence of the sometimes large day-to-

## Difference in proportion of female adults in each zone following TIRS

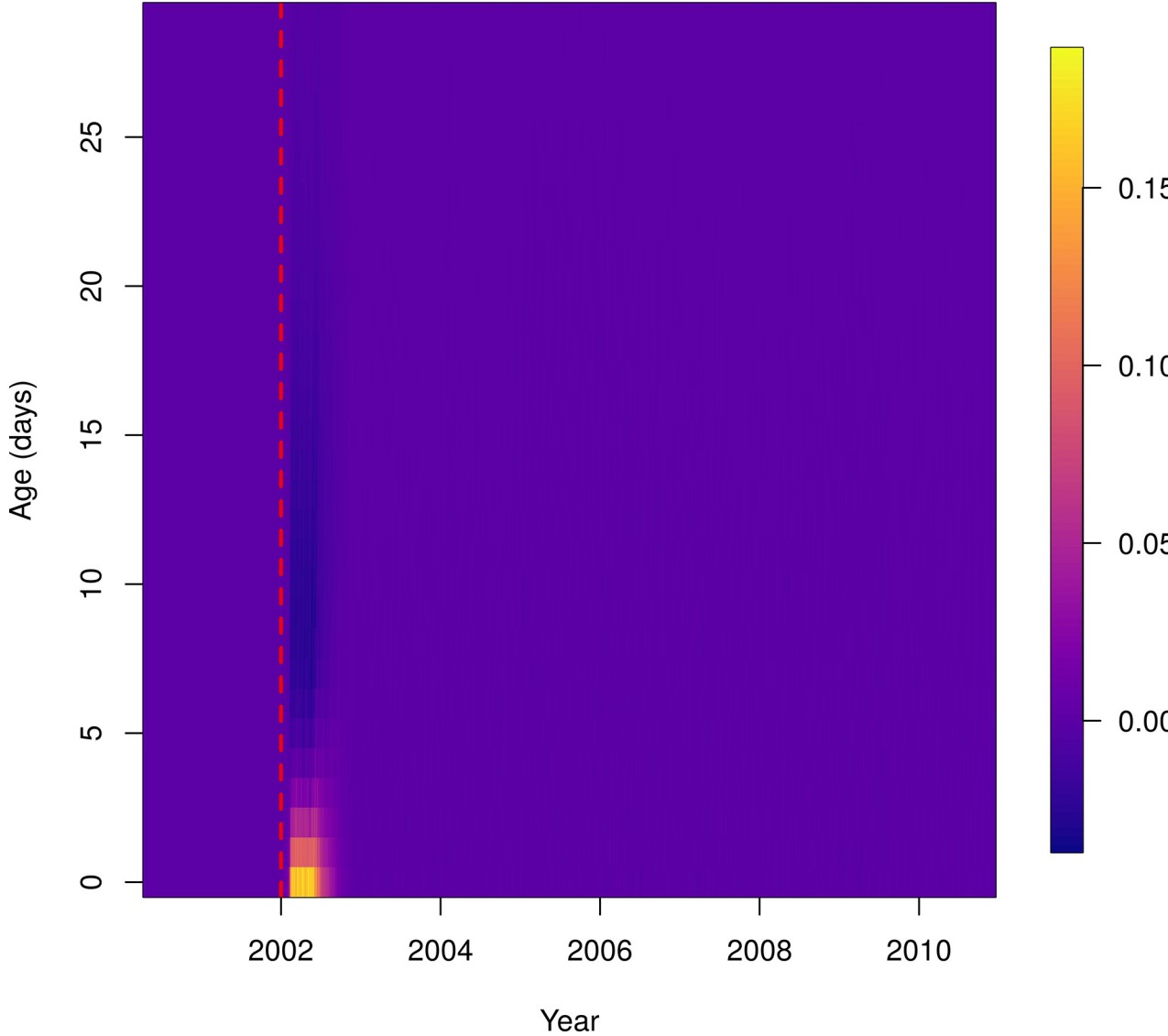

**Fig 10. Normalized difference in female adult ages over time between baseline and a city-wide TIRS campaign initiated on 1$^{st}$ January 2002, indicated by the red dashed line.** Following spraying there are proportionally more young Ae. aegypti, and this effect lasts longer than for ULV. Each column shows the difference in the daily proportion of mosquitoes of a given age, shown on the y-axis. Warmer colors represent higher proportions.

day fluctuations in abundance predicted by the GAM being incommensurate with the slower population dynamics described by the mechanistic model. The GAM can accommodate these larger fluctuations as the included environmental predictor variables vary substantially from day-to-day. It is also worth noting that the fact that our mechanistic model cannot recreate the full extent of the GAM's variability may indicate that some of these larger day-to-day fluctations may not be actually physically possible. In particular, large changes in the environmental variables could lead to large changes in the abundance predicted by the statistical model, but that were not reflected in reality. Alternatively, the fact that the mechanistic model cannot

recreate the GAM's variability could suggest there are additional processes occurring which are not able to captured by $\mu_c(t)$.

Two of the most detailed and well-established models of *Ae. aegypti* dynamics are Skeeter-Buster and AedesBA [10,23,30]. Most of the development and mortality rates in our model were based on the same formulae used in each of these models, and so results will be identical given the same environmental conditions. All three models are stochastic and spatially explicit, although our model is only fully stochastic in the adult stage. Some key differences between the models include the treatment of density-dependent larval mortality and additional mortality in the larval and pupal stages. In terms of density dependence, our model is most similar to AedesBA, in that density dependence is incorporated directly as a function of larval population size. Our model, however, includes a higher exponent in the density-dependent term (i.e., the term $\frac{L^2}{\kappa}$ in Eq 2), meaning stronger density dependence. This stronger density-dependence enables us to better match the response to spraying seen in an observational study in Iquitos by Gunning et al. [27], as a higher exponent will make the population rebound more quickly following a perturbation. Following simulation of a city-wide ULV campaign, modeled abundance returned to 10% of baseline 36 days after the culmination of ULV spraying, which was slightly slower than, but in line with, the return to baseline observed by Gunning et al., where it rebounded in a month [27]. Skeeter Buster takes a different approach and includes density dependence indirectly through competition for finite resources. This approach leads to a more oscillating return to baseline following spraying, as compared to the smoother and slower return seen in our model and AedesBA [30]. Our model's inclusion of an additional mortality term enabled us to closely match abundance estimates of a statistical model. This meant our model was better equipped to explore perturbations to actual observed mosquito abundance than Skeeter-Buster and AedesBA, but is unable to make projections of mosquito population dynamics outside that observed. Another alternative to our approach would have been to directly calibrate an agent-based population dynamics model to the household mosquito survey from Iquitos using asequential Monte Carlo approach [31] or approximate Bayesian computation [32]. The former approach can be computationally very costly for a complex model such as this, and our approach was comparatively simpler. Moreover, by fusing mechanistic and statistical models, our approach was able to leverage data on environmental covariates that may have been difficult to incorporate in a fully-mechanistic framework given the number of parameters and complex functional relationships that might entail.

A limitation of our model is that we model all immature stages aggregated at a household level, and do not model individual containers. A consequence of this is that larval competition for resources is not modeled directly. Instead it is incorporated indirectly through the density-dependent term in the equation governing the household number of larvae. This term was able to control the population size, however, and produce a realistic response to spraying by determining how the population rebounds following spraying. A related limitation is that we do not model rainfall in the agent-based model. We are, therefore, unable to directly capture any dynamical effects of changes in rainfall amounts, such as whether rainfall affects the response following spraying. Rainfall was, however, included in the statistical model to which we matched our model, so the effects of historical rainfall will be captured this way through our additional calibrated mortality. Another limitation is that the statistical model to which our method is matched is dependent on abundance surveys, which may not be available in many settings.

A strength of our approach is that it blends the strengths of statistical and mechanistic models. The underpinning statistical model means the full model has an accurate description of baseline abundance, while the agent-based model permits a dynamic response to vector

control that closely matched the population-level response observed empirically. The two-step approach and the relatively simple, deterministic treatment of immature mosquitoes, means that our model would be easy to parameterize and transfer to other settings where mosquito abundance time series are available. Because the agent-based model is also one part of a larger epidemiological model, our framework enables insight into questions of public health relevance. Because our model was able to closely recreate the abundance patterns in Iquitos, as well as produce a response to insecticide applications consistent with empirical studies [27,33–35], it provides an excellent environment to explore the impact of such control strategies on the dynamics of dengue virus transmission and disease.

## Supporting information

**S1 Fig. The time series of the estimated additional mortality parameter, $\mu_c$.** This parameter forms a key step in our approach, linking the mechanistic and statistical models, and can be thought of as accoutning for other sources of mortality that are not captured by the temperature-mortality relationships. The parameter is always positive (i.e. it is always a mortality rate). (TIF)

**S2 Fig. Time series of city-wide mosquito abundance predicted by the ABM, showing mean and 95% CI of 400 runs.** Due to the lack of stochasticity in the mosquito component of the model, there is little variability between runs, and so the 95% CI is indistinguishable from the mean. (TIF)

**S3 Fig. Change in the mortality rate following spraying.** In this example, there was one targeted insecticide residual spraying (TIRS) campaign, which began on day 10, three ultra-low volume (ULV) campaigns, or neither, which began on days 10, 17, and 24. The mortality rate increases by the shown amount. (TIF)

**S4 Fig. Mosquito abundance over time.** As in Fig 4, but not normalized by the total population (i.e., the total zonal abundance). Each column represents the daily abundance every 100 days from 2000–2010. Each row is a Ministry of Health zone in Iquitos. Columns are normalized by the total abundance that day. (TIF)

**S5 Fig. Mosquito abundance over time following a city-wide ultra-low volume spraying campaign at the start of 2002.** As in Fig 6, but not normalized by the total population (i.e., the total zonal abundance). Each column represents the daily abundance every 100 days from 2000–2010. Each row is a Ministry of Health zone in Iquitos. Columns are normalized by the total abundance that day. (TIF)

**S6 Fig. Mosquito abundance over time following a targeted indoor residual spraying campaign at the start of 2002.** As in Fig 7, but not normalized by the total population (i.e., the total zonal abundance). Each column represents the daily abundance every 100 days from 2000–2010. Each row is a Ministry of Health zone in Iquitos. Columns are normalized by the total abundance that day. (TIF)

**S7 Fig. As in Fig 6, but with all buildings set to have the same baseline abundance as the building with highest abundance.** Normalized mosquito abundance over time, following a

city-wide TIRS campaign initiated on the 1st January 2002, indicated by the red dashed line. Each column represents the daily abundance every 100 days from 2000–2010. Each row is a Ministry of Health zone in Iquitos. Columns are normalized by the total abundance that day. (TIF)

**S8 Fig. The normalized abundance by zone following A.** a city-wide campaign with a hypothetical insecticide that causes a large increase in mortality (equivalent to TIRS) with low residuality (equivalent to ULV) and B. city-wide campaign with a hypothetical insecticide that causes a small increase in mortality (equivalent to ULV) with high residuality (equivalent to TIRS).
(TIF)

**S9 Fig. Moments of the mosquito age distribution through time.** In early 2004, a cohort of mosquitoes reaches an older age, increasing all of the moments as the average age increased and the distribution becomes more skewed and bimodal.
(TIF)

**S10 Fig. The bimodality coefficient of the mosquito age distribution through time.** The bimodality coefficient takes a value between 0 and 1, and higher values mean the distribution is 'more' bimodal. It is defined as $\frac{\gamma^2+1}{\kappa}$, where $\gamma$ is the skewness and $\kappa$ is the kurtosis. In early 2004, a cohort of mosquitoes reached an older age, resulting in a bimodal age distribution.
(TIF)

**S11 Fig. Temperature time series for daily means, maxima, and minima (columns) in both air and water (rows) in Iquitos in the period 2000–2010.**
(TIF)

**S1 Text. Description of model parameterization.**
(DOCX)

**S2 Text. Outline of agent-based model.**
(DOCX)

**S1 Video. Spatial distribution following targeted indoor residual spraying.**
(MP4)

**S2 Video. Age distribution over time in the absence of spraying.**
(MP4)

**S3 Video. Age distribution over time following ultra-low volume spraying.**
(MP4)

**S4 Video. Age distribution over time following targeted indoor residual spraying.**
(MP4)

## Acknowledgments

We thank the residents of Iquitos for their participation in this study. We greatly appreciate the support of the Loreto Regional Health Department, including Drs. Hugo Rodriguez-Ferruci, Christian Carey, Carlos Alvarez, Hernan Silva, and Lic. Wilma Casanova Rojas who all facilitated our work in Iquitos. We thank the NAMRU-6 Virology and Emerging Infections Department (VEID) and Entomology Department leadership who provided institutional support, IRB guidance and support supervising field staff during the years 2000–2010 when the data used in these models was collected. We also appreciate the careful commentary and advice

provided by the NAMRU-6 IRB and Research Administration Program for the duration of this study. We thank the NAMRU-6 VEID field teams provided who daily support through duration of the project and without whom the capture of acute dengue cases would not have been possible. In particular we thank Gabriela Vasquez de la Torre for her administrative support for the project.

## Disclaimer

The views expressed in this article are those of the authors and do not necessarily reflect the official policies or positions of the Department of the Navy, Department of Defense, nor the U.S. Government.

## Copyright statement

ACM and HA were employees of the United States government. This work was prepared as part of their official duties. Title 17 U.S.C. 105 provides that 'copyright protection under this title is not available for any work of the United States Government.' Title 17 U.S.C. 101 defines a U.S. Government work as work prepared by a military service member or employee of the U.S. Government as part of that person's official duties.

## Author Contributions

**Conceptualization:** Sean M. Cavany, Alun L. Lloyd, Gonzalo M. Vazquez-Prokopec, Thomas W. Scott, Robert C. Reiner, Jr, T. Alex Perkins.

**Data curation:** Helvio Astete, Amy C. Morrison.

**Formal analysis:** Sean M. Cavany, Guido España, Alun L. Lloyd, Lance A. Waller, Uriel Kitron, Thomas W. Scott, Amy C. Morrison, Robert C. Reiner, Jr, T. Alex Perkins.

**Funding acquisition:** T. Alex Perkins.

**Investigation:** Sean M. Cavany, Guido España, Alun L. Lloyd, T. Alex Perkins.

**Methodology:** Sean M. Cavany, Guido España, Alun L. Lloyd, T. Alex Perkins.

**Project administration:** Gonzalo M. Vazquez-Prokopec, Thomas W. Scott, T. Alex Perkins.

**Resources:** T. Alex Perkins.

**Software:** Sean M. Cavany, Guido España, Robert C. Reiner, Jr, T. Alex Perkins.

**Supervision:** Guido España, T. Alex Perkins.

**Validation:** Sean M. Cavany.

**Visualization:** Sean M. Cavany, T. Alex Perkins.

**Writing – original draft:** Sean M. Cavany.

**Writing – review & editing:** Sean M. Cavany, Guido España, Alun L. Lloyd, Gonzalo M. Vazquez-Prokopec, Helvio Astete, Lance A. Waller, Uriel Kitron, Thomas W. Scott, Amy C. Morrison, Robert C. Reiner, Jr, T. Alex Perkins.

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
