## [Decision Letter · Decision Letter 0]

12 Sep 2022

Dear Dr. Cavany,

Thank you very much for submitting your manuscript "Fusing an agent-based model of mosquito population dynamics with a statistical reconstruction of spatio-temporal abundance patterns" for consideration at PLOS Computational Biology.

As with all papers reviewed by the journal, your manuscript was reviewed by members of the editorial board and by several independent reviewers. In light of the reviews (below this email), we would like to invite the resubmission of a significantly-revised version that takes into account the reviewers' comments.

While the reviewers agreed on the value, novelty and interest of the work, all three reviewers agreed that the methodology was not completely or clearly explained nor all appropriate model validation performed. This made the manuscript difficult to accurately review for a final decision. The reviewers made other important comments likely to improve the clarity and robustness of the study. Given the nature of this work as presenting a novel and potentially valuable method for parameterising agent based models of mosquito populations it is important that all methodological detail and methodology validation be provided by the authors. Thus, in their revised manuscript the authors should focus on providing complete and clear methodological details as per the reviewers comments (especially those of Reviewer 2 and 3), along with appropriate figures and supplementary results that are necessary for evaluation of the methodology and results, such as a plot of the fitted mu_c(t) parameters which appears central in the methodological flow. To this end, it would also be beneficial for review, if the authors make the accompanying data and code available to reviewers in a resubmission.

We cannot make any decision about publication until we have seen the revised manuscript and your response to all of the reviewers' comments. Your revised manuscript is also likely to be sent to reviewers for further evaluation.

Sincerely,

David S. Khoury

Academic Editor

PLOS Computational Biology

Thomas Leitner

Section Editor

PLOS Computational Biology

Reviewer's Responses to Questions

**Comments to the Authors:**

Reviewer #1: This manuscript by Cavany et al. reports a new model of Aedes aegypti mosquito population dynamics and its application to vector control. The novelty of the authors’ approach is that they use statistically derived estimates of mosquito abundance from survey data and use these estimates to drive the baseline dynamics of a stochastic agent-based model. This latter model, in turn, can be applied to simulate the effects of perturbations to the population – in this case, comparing the effects of two different insecticide spraying regimens across space and time. Overall, this is a strong manuscript that presents a significant advance in vector modeling research, namely, a model that retains faithfulness to empirical data via statistical estimates, while providing predictive power to simulate the effects of interventions or environmental perturbations. My sole criticism is for the authors’ to perform more exploration of the parameter space on the effects of ULV and TIRS, to determine how robust their results are across a range of potential parameter values:

pg. 14: The authors describe how the parameters governing the effects of ULV or TIRS were calibrated to existing data, however it is helpful to demonstrate how robust the results are to variations in these parameter values that may be seen in real-life conditions that do not conform exactly to the cited studies. I recommend the authors perform an exploration of the parameter space by (1) varying the adult mortality rate from ULV and (2) the adult mortality rate and duration of effect from TIRS within a realistic range of values and report how these parameter variations affect mosquito abundance both overall as well as spatially by MOH zone.

Additional strengths of the manuscript include the clarity of the writing and data presentation.

Reviewer #2: Please see my detailed comments in the attached pdf.

Reviewer #3: In “Fusing an agent-based model of mosquito population dynamics with a statistical reconstruction of spatio-temporal abundance patterns”, Cavany et al present a novel method for parameterizing an agent based model of mosquito dynamics based on household survey data from Iquitos, Peru. While their method requires a number of intermediate steps that are not always trivial, evident by their first step having been published on its own, it is likely still preferable to current methods of parameterizing ABMs, e.g. approximate-bayesian computation. Overall, the paper is well written and easy to follow. However, there are some minor comments that I believe should be addressed before publication.

• In the methods for the deterministic model, I am confused by the third and final steps. µc(t) is not in equation 4 and why would L(t) not be obtained from equation 2 (d(L(t))/dt)? I am not sure if this is simply a numbering issue or if I am missing something, but it needs clarification.

• In the methods for the experiments, I am not sure what to make of the increased mortality due to ULV and residual spraying. Typically, mortality rates have a /day unit. Does this mean that there are 1.5 (or 9) additional deaths per day regardless of population size? If so, what is done to keep population sizes above 0? Or is this perhaps the number of additional deaths assuming the mean or equilibrium population size?

• What is the justification for applying the residual spray to every household in the city? This does not seem like a realistic choice. At the bare minimum, the authors should discuss the number of households that could be reasonably expected to be treated in a given period.

• When comparing the ABM to the ODE and GAM, the authors should mention how many runs of the ABM are being averaged over. Related to this, is the trajectory given for the ABM in figure 2 a mean trajectory or from an individual run? It would also be helpful to include some mention of how much variation is seen between model runs.

• If the trajectory in Figure 2 is from an individual run, it seems that the ABM, as well as the ODE, end up smoothing out much of the variation that is apparent in the GAM. This would be fine if that variation is not biologically relevant or representative of actual changes in the population, e.g. measurement error, but I do not expect this is the case. I think this is something the authors should discuss. If the trajectory shown is in fact a mean trajectory, it would be beneficial to know how a single run compares to the GAM.

• In general, the manuscript would benefit from expanded figure captions that include the take-away message. For some of the figures (i.e. Figure 6), I am not sure what I am supposed to see.

• In the results on the mosquito age distribution following spraying. I am confused about the last sentence of this paragraph, “Occasionally, such as near the start of 2004, a cohort of adult mosquitoes survived longer and the age distribution became less skewed and sometimes bimodal”. What is happening here? Is this just a result of the stochasticity of the model at low population sizes? Does it show up across multiple simulations or is this a result that is only seen in a single or a few simulations. Again, it would be good to know if these results are from several simulation runs or only a single simulation.

• The authors should include a justification for why only Tmax is used for estimating mortality rates. How would using another measure affect the results?

• Since the thermal response curves used are justified based on the fact “the temperature never gets cold enough to cause mortality from cold temperatures”, the authors should include at least summary information on the temperatures observed in Iquitos. The authors should also justify the use of Magori et al and Otero et al over more recent publications, e.g. Mordecai et al. 2017, which estimated thermal response curves for many of these parameters based on experimental data. Especially when the thermal responses presented in Mordecai et al would not necessitate the temperature not falling below the optimal temperature for mosquito mortality, which I doubt.

• Figure S1, what is the unit on change in mortality rate? Is this an absolute change (/day) or a relative change?

• The authors should discuss how this method compares to other ways of calibrating ABM to this type of data.

Typos and miscellaneous:

Abstract: yellow virus should be yellow fever virus

Figure 2: The line is described as blue. It appears to be purple. Is the fact that the trajectory lines extend beyond the x-axis an intentional choice?

Figure 5: the y-axis title and label text are overlapping

**Have the authors made all data and (if applicable) computational code underlying the findings in their manuscript fully available?**

Reviewer #1: Yes

Reviewer #2: **No: **It is noted by the authors that "All necessary computer code will be made available on GitHub prior to publication", but I did not receive a copy of any code or data for this review, nor a link to the relevant GitHub repository.

Reviewer #3: **No: **Authors do not make code available as part of this manuscript. However, most if not all of it should be available as part of previous publications. That being said, for someone attempting to replicate their methods it would be useful to have code that details how all the parts are integrated.

PLOS authors have the option to publish the peer review history of their article (what does this mean?). If published, this will include your full peer review and any attached files.

Reviewer #1: **Yes: **Joshua R. Lacsina

Reviewer #2: No

Reviewer #3: No
---

## [Decision Letter · Decision Letter 1]

29 Mar 2023

Dear Dr. Cavany,

Thank you very much for submitting your manuscript "Fusing an agent-based model of mosquito population dynamics with a statistical reconstruction of spatio-temporal abundance patterns" for consideration at PLOS Computational Biology. We are pleased to inform you that the reviewers agreed the manuscript is much improved by the authors latest revisions. The manuscript is thus able to be accepted, but would first ask that you consider the recommended minor changes/corrections suggested by the reviewers. 

Sincerely,

David S. Khoury

Academic Editor

PLOS Computational Biology

Thomas Leitner

Section Editor

PLOS Computational Biology

Reviewer's Responses to Questions

**Comments to the Authors:**

Reviewer #1: The authors have fully addressed all my critiques.

Reviewer #2: Please see my detailed comments in the attached PDF.

Reviewer #3: The manuscript “Fusing an agent-based model of mosquito population dynamics with a statistical reconstruction of spatiotemporal abundance patterns” has been much improved following revisions and the authors have addressed all of my concerns. The paper is now very clearly written and the methods are easily understood. My opinion is that the methods described in this manuscript will be useful for the development of future models of mosquito control. I have a few very minor comments that I believe further improve the manuscript.

• On line 89, the authors describe estimating population abundance as being “typically” done based on MMR. While I agree that it is often done this way, I feel that it more often done based on trap data due to the ease and availability of trap data.

• On lines 335-336 the authors state “This is likely a consequence of the precipitous drop in abundance around this time necessitating a large value of μc(t).” Do the authors believe this to be a numeric relic from the fitting process or reflective of something biological? If it is biological, do the authors have any theories as to what caused it and its likely impacts?

• On lines 383-385 the authors make the comment “It is also worth noting that the fact that our mechanistic model cannot recreate the full extent of the GAM’s variability may indicate that some of these larger day-to-day fluctuations may not be physically possible.” I think this is an extremely good point that the authors could expand further on. Also fluctuations is misspelled in the text.

**Have the authors made all data and (if applicable) computational code underlying the findings in their manuscript fully available?**

Reviewer #1: Yes

Reviewer #2: Yes

Reviewer #3: Yes

PLOS authors have the option to publish the peer review history of their article (what does this mean?). If published, this will include your full peer review and any attached files.

Reviewer #1: **Yes: **Joshua R. Lacsina

Reviewer #2: No

Reviewer #3: No

Figure Files:

Data Requirements:

Reproducibility:

References:

---

## [Editor Report · Decision Letter 2]

11 Apr 2023

Dear Dr. Cavany,

We are pleased to inform you that your manuscript 'Fusing an agent-based model of mosquito population dynamics with a statistical reconstruction of spatio-temporal abundance patterns' has been provisionally accepted for publication in PLOS Computational Biology.

Best regards,

David S. Khoury

Academic Editor

PLOS Computational Biology

Thomas Leitner

Section Editor

PLOS Computational Biology

---

## [Editor Report · Acceptance letter]

24 Apr 2023

PCOMPBIOL-D-22-01121R2 

Fusing an agent-based model of mosquito population dynamics with a statistical reconstruction of spatio-temporal abundance patterns

Dear Dr Cavany,

I am pleased to inform you that your manuscript has been formally accepted for publication in PLOS Computational Biology. Your manuscript is now with our production department and you will be notified of the publication date in due course.

With kind regards,

Anita Estes
